# Intimate partner violence and unmet need for family planning in the Democratic Republic of the Congo: A secondary analysis of the moderating role of reproductive coercion using performance monitoring for action data

Didier Mbombo Ndombe[1,2,3]☯*, Ester Lucia Rizzi[1]☯

**1** Centre for Demographic Research, Université catholique de Louvain, Louvain-la-Neuve, Belgium, **2** Centre Interdisciplinaire pour le Développement et l'Éducation Permanente, Kinshasa, Democratic Republic of the Congo, **3** Ecole des Sciences de la Population et du Développement, Université de Kinshasa, Kinshasa, Democratic Republic of the Congo

☯ These authors contributed equally to this work.
* didier.ndombe@uclouvain.be

## Abstract

### Introduction

Intimate partner violence (IPV) has harmful effects on women's physical, mental, and reproductive health. This study investigates the relationship between experiencing IPV and unmet need for family planning in two provinces of the Democratic Republic of the Congo (DRC), emphasizing the potential mediating effect of reproductive coercion. To our knowledge, this is the first examination of the connection between IPV and unmet need for family planning in the DRC, particularly regarding the underlying mechanisms of this association.

### Materials and methods

This research utilized secondary data from the third wave of the Performance Monitoring for Action (PMA) surveys carried out in two provinces of the DRC during the period from 2021 to 2022. The analysis focused on a sample comprising 1,387 women in unions aged between 15 and 49 years. To investigate the relationship between IPV and unmet need for family planning, linear probability models were employed and various factors were controlled. Additionally, we explored the moderating influence of reproductive coercion in this context.

### Results

The linear probability model indicated that experiencing sexual IPV within the past 12 months was linked to an unmet need for family planning [aCoef = 0.12, CI95% =

**Data availability statement:** All data files are available from the IPUMS PMA database (https://pma.ipums.org/pma/) or directly via the PMA database using the following accession number: https://doi.org/10.34976/dgpr-sz30.

**Funding:** The Université catholique de Louvain (Belgium) provided funding for this PhD research [http://dx.doi.org/10.13039/100007370 / Didier Mbombo Ndombe].

**Competing interests:** The authors have declared that no competing interests exist.

0.04–0.21]. When sexual IPV occurred alongside reproductive coercion, a stronger correlation with unmet need was detected [aCoef = 0.24, CI95% = 0.03–0.44].

## Conclusion

We showed that sexual IPV is associated with an unmet need for family planning in the DRC. While IPV should be considered a policy target, these results emphasize the importance of responding appropriately to IPV when sexual and reproductive health services are provided to women. This was the first study on the DRC and, more generally, on sub-Saharan Africa (SSA) to illuminate the underlying mechanisms of the association, revealing the moderating role of reproductive coercion.

## Introduction

The term "unmet need for family planning" refers to the absence of contraceptive use among sexually active women of reproductive age who express a desire either to avoid having additional children or to postpone the arrival of their next child for a minimum period of two years [1]. Worldwide, 257 million women of reproductive age still have an unmet need for modern contraception [2], and three quarters live in low- and middle-income countries (LMICs) [3]. In sub-Saharan Africa (SSA), which is characterized by low contraceptive usage rates [4], the estimated prevalence rate of unmet need for family planning is 23.7%. This study includes 15.8% of women seeking to space their births and 7.9% aiming to limit the number of births [5].

Unmet reproductive health needs are linked to increased fertility rates, preterm birth, fetal mortality, low birth weight, neonatal death, and unintended pregnancies [3]. In LMICs, the annual occurrence of unintended pregnancies is projected to be approximately 74 million; this situation can result in approximately 25 million abortions conducted under unsafe conditions [6], thereby heightening the risk of maternal mortality. In regions such as Latin America and Africa, it is estimated that 75% of abortions are performed unsafely [7].

The exploration of the unmet need for family planning is often based on Bronfenbrenner's *ecological systems theory.* The theory suggests that women's contraceptive practices are influenced not only by their personal attributes but also by the larger social environment in which they reside, encompassing both their household and community dynamics [8,9]. Consistent with this theoretical framework, recent studies conducted in LMICs have identified a range of factors linked to the unmet need for family planning services. These factors include age, knowledge of contemporary contraceptive methods [10], exposure to mass media [11], educational attainment, length of cohabitation, parity [12], employment status [13], and religion [14]. Other authors have highlighted the impact of household wealth quintiles [11], the importance of discussions between partners [15], and the role of women's decision-making power in health care [5]. Additionally, some studies have emphasized the influence of contextual factors, including gender inequality at the contextual level [16] and place of residence [17], although a lack of agreement regarding the influence of the latter

remains. Research has shown that intimate partner violence (IPV) considerably affects the family planning requirements of victims [18–21]. This effect is also the primary focus of our study.

With respect to the issue of IPV, it is essential to recognize the various forms that may be displayed. These manifestations encompass physical, sexual, and emotional IPV in addition to controlling behaviors exhibited by the intimate partner or spouse [22]. In 2018, the study reported that more than a quarter (27%) of women worldwide who have ever been married or in a partnership have encountered violence—either physical or sexual—at least once in their lifetimes from an intimate partner, beginning from the age of 15 years [1]. Globally, the prevalence of such IPV in the past 12 months is estimated at 13%. IPV presents a pervasive concern worldwide, affecting all communities [23], with a notably higher prevalence observed in low-income countries [24]. In SSA, the prevalence of physical or sexual IPV is estimated to be 30% [25]. An abundance of research links the consequences of IPV to both long-term health issues [26–28] and considerable financial losses [29], such as expenses related to medical care, diminished productivity among victims, and costs associated with separation.

IPV might reduce victims' reproductive decision-making power [30,31] in terms of contraceptive methods, pregnancy and motherhood [18,32]. From an alternative perspective, women who have been victims of IPV may be more inclined to use contraception to avoid pregnancy and to prevent the birth of a child in a violent environment [19]. At the empirical level, previous quantitative studies have yielded divergent results regarding the association between IPV and reproductive health outcomes, such as unmet need for family planning, unwanted pregnancies and nonuse of contraception [22,33–36]. A positive association between IPV and unmet need for family planning has been reported in SSA according to pooled data from 26 countries [18] as well as in Pakistan [20], Colombia [31], Estonia [37], and Jordan [38]. An inverse effect of IPV on unmet need for family planning use was observed in a study conducted in SSA limited to six countries [22], as well as in Afghanistan [19] and three South Asian countries [39]. No significant association was found between IPV and unmet need nor between the use of family planning or contraceptive methods in either Maldives [20] or Philippines [40,41].

Few studies consider the mechanisms by which IPV leads to unmet need. The quantitative study by Miller [42] in the context of northern California is one of these exceptions. Using a multivariate stratified analysis, Miller shows the combined effect of IPV and reproductive control on unintended pregnancy. Other studies also assume that violent men may use coercive tactics to induce pregnancy and direct interference with contraceptive methods, although only at the theoretical level [39]. In addition, qualitative studies show that women who are victims of IPV may be less inclined to use contraception for fear of further violence perpetrated by their partners [43] or because they are unable to negotiate the use of family planning with violent partners [15], which also encompasses a context of reproductive coercion.

Although some studies have failed to offer empirical evidence for the occurrence of reproductive coercion perpetrated by men [18], other investigations suggest that men might exert control over women's reproductive choices through coercive behaviors, such as sabotaging the use of contraceptives and pressuring women to become pregnant [43–45]. Women experiencing physical and sexual IPV are particularly vulnerable to reproductive control [42,43], especially in LMICs. Other risk factors for reproductive coercion include cohabitation without marriage, polygyny, low household income, and a small number of children [44]. Furthermore, qualitative studies have shown that social norms that favor procreation and gender inequality impact reproductive health [46].

Research examining the interplay between reproductive coercion and IPV in LMICs and SSA has been quite limited, primarily because of a scarcity of available data [32,44]. In particular, studies focusing on IPV and reproductive coercion are lacking within the DRC [47] despite high rates of IPV in the country (44% in 2014) and despite equally high unmet need for family planning, with an increasing trend from 28.7% in 2018 to 31.6% in 2024 [48–50].

To the best of our knowledge, this study is the first to examine the relationship between IPV and unmet need for family planning specifically in the DRC, as previous studies on SSA considered pooled data [18] or countries other than the DRC [22]. Moreover, to our knowledge, this is the first study to empirically consider reproductive coercion as a modifier of this association in the DRC and, more generally, in SSA. We formulated two main hypotheses. In line with the previous

literature—particularly studies that pool together a large number of SSA countries—in our first hypothesis (H1), we postulated that in the DRC, IPV has a positive effect on unmet need for family planning. Women who experience IPV are more likely to have an unmet need for family planning than those who have not. With respect to the second hypothesis (H2), we aimed to empirically test the reproductive coercion hypothesis only, which was theoretically proposed by Raj and McDougal [39], and we postulated that reproductive coercion interacts with IPV to increase the risk of unmet need for family planning.

## Materials and methods

### Data source

We used secondary data from the Performance Monitoring for Action (PMA) project database. The project was implemented with the financial support of the Bill and Melinda Gates Foundation and received ethical approval [51]. The PMA is a research platform that administers a longitudinal survey model with integrated cross-sectional surveys of households, women and institutions in eight countries in SSA and Asia [51]. The PMA uses an open, prospective, observational cohort approach to analyze trends in contraceptive use and changes in women's fertility intentions over time.

The dataset utilized in this analysis is particularly suitable because it encompasses a diverse range of variables, exceeding the breadth of any other national survey available in the DRC. For example, the PMA offers a more detailed assessment of reproductive coercion through several items, whereas the Demographic and Health Surveys (DHS) limit their inquiry to a single question concerning coercion related to pregnancy.

The PMA employs a multilevel stratified cluster sampling design, using urban–rural strata and/or large regions as the sampling units, similar to the approach taken in the DRC. The sampling units are geographical clusters listed by the national statistical office. Clusters are drawn using probability proportional to size methods. Moreover, unlike other countries, the DRC data are not representative at the national level but rather at the subnational level. The sampling technique was designed to provide province-level estimates of the modern contraceptive prevalence rate among all women with a margin of error of 5%. In each cluster, 35 households were selected randomly for a face-to-face interview with all eligible women aged 15–49 years in the households. The full details of the methodology were published by the PMA [51].

For the purposes of this study, the analytical sample included women in unions residing in the DRC, specifically in the provinces of Kinshasa and Kongo Central, where the PMA project was being implemented. Kinshasa is a densely populated, predominantly urban capital with relatively better access to health services and information, while Kongo Central is distinguished by a substantial predominance of rural areas and faces significant inequalities in access to essential resources and services. With the objective of examining the association between intimate partner violence (IPV) and unmet need for family planning, the study population was restricted to women aged 15–49 years living in a union, that is, women who are at risk of pregnancy or presenting reproductive potential. In addition, the analyses were limited to women in unions who responded to the module on violence against women and girls to capture information on IPV. All other individuals were excluded from the analytical sample using the "marital status" variable. In summary, among the 2,621 women in unions who participated in the third phase of the survey (Panel 2022), only 1,387 (52.9%) satisfied the predefined inclusion criteria. After the sample was restricted to observations with complete information on all the variables used in the statistical models, 1,382 eligible women were retained for the subsequent analyses. The corresponding sampling flow diagram is presented in S1 Fig.

In this study, we used data from the most recent phase (Phase 3) of 2022 and thus adopted a cross-sectional approach. This choice is justified by the fact that information on IPV (our independent variable) was collected only during this phase.

### Description of variables

**Dependent variable.** An unmet need for family planning is the dependent variable. It is a constructed measure commonly used in research that assesses a woman's need for family planning on the basis of her fertility preferences, current use

of family planning methods, and risk factors for pregnancy. This measure of unmet need is a binary variable (Yes, No) and is constructed by summing the two components of unmet need, namely, unmet need for spacing and unmet need for limiting. *Unmet spacing needs* are (i) pregnant women whose pregnancy was mistimed; (ii) women in postpartum amenorrhea whose last birth was mistimed; and (iii) fertile women who are neither pregnant nor in postpartum amenorrhea, who are not using any family planning method, who say they want to wait two years or more before their next birth, who are undecided about the timing of their next birth, or who are undecided about having another child. The *unmet limitation need* includes the following: (iv) pregnant women whose pregnancies were unwanted; (v) women in postpartum amenorrhea whose last birth was unwanted; and (vi) fertile women who are neither pregnant nor in postpartum amenorrhea, who do not use any family planning methods and who do not want any more children [51].

**Main independent variable**. The primary independent variable in this study is IPV. The evaluation of IPV was conducted using the Conflict Tactics Scale (CTS) [52]. The following questions were asked: "In the past 12 months, has your husband/partner (a) insulted, reprimanded, shouted at or made derogatory comments about you; (b) slapped, hit or otherwise physically abused you; (c) threatened you with a weapon or tried to strangle or kill you; (d) pressured or insisted on having sex with you when you did not want to (without using physical force); or (e) physically forced you to engage in sexual intercourse?" These questions led to the creation of three main variables of IPV: sexual IPV (questions d and e), physical IPV (questions b and c), and emotional IPV (question a). A dichotomous variable was created for each of these three variables, coded 1 for 'Yes' and 0 for 'No.' In addition, the scores for the three components (sexual, physical and emotional) were added to construct a global variable of IPV, also dichotomous, coded 1 for 'Yes' and 0 for 'No'. A score of zero, coded 0, indicated that the woman had not been subjected to any form of IPV during the past 12 months. A score greater than or equal to 1, coded 1, indicated that the respondent had been a victim of at least one form of IPV during the 12 months preceding the survey.

**Interaction variable**. In this study, reproductive coercion is identified as a potential interaction factor. The assessment of reproductive coercion comprises multiple items that capture coercive behaviors associated with both pregnancy and contraception use: "In the past 12 months, has your husband/spouse (a) made you feel bad or treated you badly for wanting to use a family planning method to delay or avoid pregnancy; (b) tried to force or push you to get pregnant; (c) told you that he would leave you if you did not get pregnant; (d) told you that he would have a child with someone else if you did not get pregnant; or (e) confiscated your family planning method or prevented you from going to the clinic to get a family planning method" [51].

Each of these questions was coded 1 for 'Yes' and 0 for 'No'. In addition, these five variables were combined to construct the total reproductive coercion variable, which was also dichotomous, coded 1 for 'Yes' and 0 for 'No'. The scores for the five variables were added together. The variable was assigned a code of 0 when the total was equal to 0, signifying the absence of reproductive coercion within the previous 12 months. In contrast, when the total was greater than or equal to 1, the variable received a code of 1, indicating that the respondent had experienced reproductive coercion from an intimate partner during the same time frame.

**Covariates**. Furthermore, we accounted for several variables: the educational attainment of both the woman and her partner, the woman's employment status over the past 12 months, the number of children (parity), the province where she resides, the religion of the household head, the woman's age, and the household wealth tercile. This approach is supported by the existing literature that recognizes these factors as correlated with both IPV and unmet need for family planning, potentially leading to a partially spurious association.

The educational background of the couple was assessed through a combination of a woman's and her partner's education, classified into four distinct categories: "high-high", "low-low", "woman more educated than the man", and "woman less educated than the man". The variable pertaining to the woman's employment status was divided into two categories: "has worked in the past 12 months" and "has not worked in the past 12 months". Parity was categorized into four groups: "nulliparous", "one or two children", "three to four children", and "five or more children". The provinces of residence were

identified as "Kinshasa" and "Kongo Central". Religious affiliation was restructured into four categories: "no religion", "Christians", and "other religions". Age was grouped into "15 to 24 years", "25 to 34 years", and "35 to 49 years". Finally, wealth was stratified into terciles labeled "lowest", "medium", and "highest". This categorization involved recoding the original variable "wealth quintile," which included five categories. The "lowest quintile" and "second quintile" were merged to form the "lowest" category, whereas the "fourth quintile" and "fifth quintile" were consolidated into the "highest" category.

Two imputation methods were utilized to manage missing data associated with the "religion" variable and the "spouse's level of education" variable. Although we initially detected 18 (1.3%) missing values out of a total of 1,382 observations, we opted not to eliminate these cases from our analysis; rather, we introduced a "missing" category to account for them. We subsequently employed stationary completion as an imputation technique [53]. The 21 (1.5%) values identified in the "partner's educational attainment" variable were addressed by assigning them the value corresponding to the most frequently occurring category, thus enabling their seamless integration with the "woman's education attainment".

## Statistical analysis

We used Stata software version 18 to perform the statistical analysis. First, we describe the sociodemographic traits of the participants with descriptive statistics, showing both absolute (n) and relative (%) frequencies. We calculated the rates of IPV and unmet need for family planning over the past 12 months using percentages along with 95% confidence intervals (CIs). Weights were used to fix issues related to having too few or too many samples during the survey sample selection process. Following the PMA guidelines, the weighting variable "FQWEIGHT" was used to adjust the analyses of all the questions from the women's questionnaire. In addition, to evaluate the statistical relationships between the explanatory variables and unmet need for family planning, we calculated the unadjusted effects of each variable through simple linear regression analysis, applying a significance level of 5% (p < 0.05). Furthermore, linear probability models (LPMs) were employed, with unmet need for family planning serving as the dependent variable, IPV identified as the main explanatory variable, and reproductive coercion included as an interaction variable. Although the outcome variable was binary, the LPM was preferred because of its ability to remedy problems encountered with the logistic model, particularly the problem of unobserved heterogeneity, which makes comparisons of odds ratios across multiple models inappropriate [54]. We incorporated interaction terms that represent the joint effects of the various types of IPV alongside exposure to reproductive coercion.

These models were modified to account for potential confounding variables, which included the education attainment of both the woman and her partner, the women's employment status, age, parity, religion affiliation, socioeconomic status as indicated by wealth terciles, and geographic location (province of residence). The adjusted coefficients provide estimates of the relationships between the independent variables and the dependent variable. A p value threshold of 5% (p < 0.05) or less with 95% confidence intervals was considered to indicate the statistical significance. Prior to model estimation, a test for heteroskedasticity was conducted utilizing the Breusch–Pagan/Cook–Weisberg method. A rejection of the null hypothesis in this context suggests a lack of homoskedasticity within the model. Consequently, it became essential to incorporate the "robust" option at the conclusion of the linear regression command in Stata to rectify standard errors through the Eicker-White robust variances [55]. The results of the Breusch–Pagan/Cook–Weisberg test for heteroskedasticity are provided in the supplementary material (S2 Table).

## Results

### Descriptive statistics

The sociodemographic characteristics of the participants are summarized in Table 1. More than 80% of the women were aged 25 years or older. Most of the women already had at least one child, while a nonnegligible portion (5.4%) of the women were nulliparous. Approximately three-quarters of the respondents lived in high-education homogamous couples.

**Table 1. Prevalence of unmet need for family planning according to selected covariates (n = 1382).**

| Variables | N | % [a, b] | Unmet need N (%) [a, c] |
|---|---|---|---|
| **Sexual IPV** | | | |
| No | 1199 | 87.8 | 239 (19.7) |
| Yes | 183 | 12.2 | 55 (34.2) |
| **Physical IPV** | | | |
| No | 1200 | 87.8 | 244 (20.6) |
| Yes | 182 | 12.2 | 50 (27.6) |
| **Emotional IPV** | | | |
| No | 1022 | 74.2 | 222 (22.1) |
| Yes | 360 | 25.8 | 72 (19.6) |
| **Any IPV** | | | |
| No | 921 | 67.5 | 184 (20.2) |
| Yes | 461 | 32.5 | 110 (24.1) |
| **Reproductive coercion** | | | |
| No | 1217 | 88.6 | 250 (20.5) |
| Yes | 165 | 11.4 | 44 (26.7) |
| **Couple's educational attainment** | | | |
| Low – low (low-education homogamous) | 108 | 10.0 | 37 (34.7) |
| High – high (high-education homogamous) | 1031 | 74.2 | 196 (19.1) |
| Women are more educated than men (hypogamous) | 33 | 2.1 | 10 (33.5) |
| Women are less educated than men (hypergamous) | 210 | 15.6 | 51 (22.3) |
| **Religion of head of the household** | | | |
| No religion | 85 | 7.0 | 21 (25.1) |
| Christian | 986 | 70.9 | 195 (19.6) |
| Other religion | 293 | 21.0 | 74 (27.1) |
| Missing | 18 | 1.0 | 4 (13.2) |
| **Woman's employment status (last 12 months)** | | | |
| Did not work | 705 | 51.9 | 183 (26.3) |
| Worked | 677 | 48.1 | 111 (16.3) |
| **Tercile of wealth** | | | |
| The lowest | 513 | 40.0 | 132 (24.9) |
| Medium | 259 | 20.1 | 61 (23.3) |
| The highest | 610 | 39.9 | 101 (17.1) |
| **Province of residence** | | | |
| Kinshasa | 701 | 49.7 | 128 (17.3) |
| Kongo Central | 681 | 50.3 | 166 (25.7) |
| **Number of children** | | | |
| Nulliparous | 74 | 5.4 | 16 (17.9) |
| 1–2 children | 481 | 33.3 | 92 (18.5) |
| 3–4 children | 458 | 33.1 | 96 (23.6) |
| 5 children or more | 369 | 28.2 | 90 (23.2) |
| **Woman's age (group)** | | | |
| 15–24 | 234 | 15.9 | 69 (29.2) |
| 25–34 | 567 | 41.6 | 121 (22.3) |
| 35–49 | 581 | 42.5 | 104 (17.8) |

**Notes**: Cross-tabulations were performed between each covariable and unmet need for family planning. The 95% confidence intervals (CIs) are in parentheses.

[a] Weighted percentage.

[b] Distribution of the independent variable (% calculated in column).

[c] N and % of women with unmet need for family planning for each category of the independent variable (% calculated in row).

Slightly more than half (51.9%) of the women had not engaged in employment during the previous 12 months. Furthermore, 40% of the participants lived in households classified within the lowest wealth tercile. Finally, a considerable percentage (70.9%) of the women resided in households led by individuals identifying as Christian.

Overall, slightly more than 3 in 10 women (32.5% [95% CI: 29.6–35.4], Table 1) reported experiencing some form of IPV, whether physical, sexual or emotional, within the past 12 months. These findings align with the prevalence rates of IPV in SSA documented by Ouedraogo & Stenzel [25] and are similar to the recent 2023–2024 DRC DHS estimate of 33% [50] and lower than the 43.9% reported in the Demographic Health Survey 2014 for the whole DRC. An examination of the data concerning the specific types of IPV encountered by the women revealed that slightly more than one quarter (25.8%) had experienced emotional abuse. The prevalence rates for the other two forms of sexual and physical IPV were similar at approximately 12%.

With respect to reproductive coercion, slightly more than one in ten women (11.4%) [95% CI: 9.6–13.4] have been subjected to reproductive control (refer to Table 1), which includes pressure to either become pregnant and/or utilize contraception. As illustrated in Fig 1, this rate is notably higher among women who have encountered IPV within the past 12 months (20.8%) than among those who have not faced such IPV (7.5%). Additionally, as expected, significant differences were observed across various types of IPV. The incidence of reproductive control was particularly elevated among women who had experienced sexual IPV in the past 12 months (29.5%), followed by those who had experienced physical IPV (25.8%), while women who had experienced emotional abuse from an intimate partner reported a lower prevalence rate at 19.4%.

With respect to the prevalence of unmet need for family planning in the 12 months prior to the survey (see Table 2), one in five women in a union (21.5% [95% CI: 19.1–24.1]) had an unmet need for family planning, with 17.1% [95% CI: 14.9–19.5] of women having an unmet need for birth spacing and 4.4% [95% CI: 3.3–5.9] of women having an unmet need for birth limitation.

In the final column of Table 1, the prevalence of unmet need for family planning is presented for each category of all the covariates. The study revealed that the prevalence of unmet need for family planning was significantly greater among

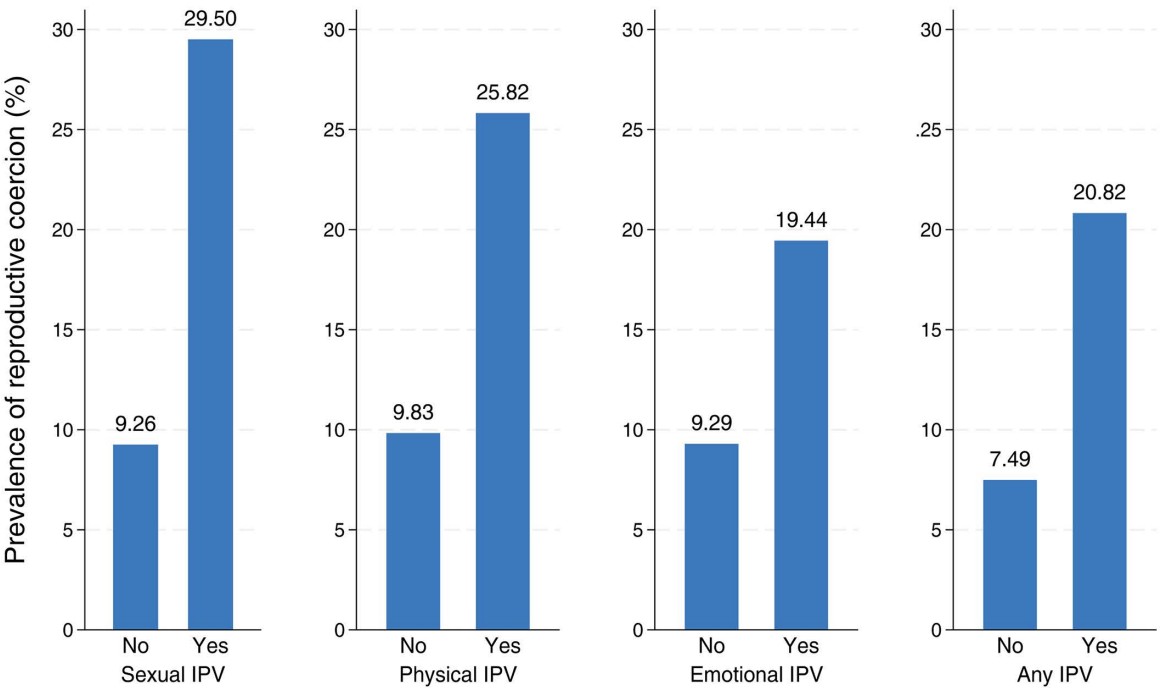

**Fig 1. Prevalence of reproductive coercion in the past 12 months by form of IPV.**

**Table 2. Unmet need for family planning among women in unions during the past 12 months (n = 1,382).**

| Variable | N | % [a] | 95% CI |
|---|---|---|---|
| **Family planning unmet need** | | | |
| Women with an unmet need for family planning (total) | 298 | 21.5 | (19.10–24.11) |
| Women with an unmet need for birth limitation | 61 | 4.4 | (3.30–5.91) |
| Women with an unmet need for birth spacing | 237 | 17.1 | (14.92–19.50) |

Notes: Tabulations were performed. The 95% confidence intervals (CIs) are in parentheses.

[a] Weighted percentage.

individuals who had experienced sexual IPV (34.2%) than among those who had not (19.7%). The prevalence of unmet need was found to be greater among victims of physical IPV (27.6%) than among those who had not experienced physical IPV (20.6%). In addition, a lower prevalence of unmet need was observed among women who had experienced emotional IPV (19.6%) than among those who had not (22.1%).

Furthermore, a higher prevalence of unmet need for family planning was observed among women living in low educational homogamy (34.7%), those with a higher level of education than their partners (33.5%), the youngest women aged 15–24 (29.2%), those who had not worked for pay in the past 12 months (26.3%), women living in a household with the lowest wealth tercile (24.9%), and those with the most children (23.6%). A higher prevalence of unmet need was observed among women of the religious denomination 'other' (27.1%) and those with no religious affiliation (25.1%) than among Christian women (19.6%). The prevalence of unmet need for family planning was also greater in the province of Kongo Central (25.7%) than in Kinshasa (17.3%).

## Multivariate models

Table 3 presents unadjusted coefficients of the associations between each form of IPV and unmet need for family planning among women in unions, together with adjusted coefficients once the woman's age, employment status, the couple's educational attainment, parity, wealth tercile, religion and province of residence are controlled (see also S3 Table).

The adjusted coefficients presented in Table 3 corroborate the descriptive findings and suggest that women who have encountered sexual IPV are more likely to experience an unmet need than those who have not [aCoef = 0.12, 95% CI = 0.04–0.21]. In contrast, no statistically significant association was found between the experience of physical IPV and unmet need for family planning [aCoef = 0.07, 95% CI = −0.01–0.15] or between emotional IPV and unmet need for family

**Table 3. Linear probability models of the association between IPV and unmet need for family planning (n = 1,382). Unadjusted and adjusted coefficients.**

| Model | Variable | Unadjusted coefficients | 95% CI | Adjusted Coefficients [a] | 95% CI |
|---|---|---|---|---|---|
| Model 1a | Sexual IPV | **0.14*** | **(0.06; 0.22)** | **0.12*** | **(0.04; 0.21)** |
| Model 1b | Physical IPV | 0.06 | (−0.01; 0.14) | 0.07 | (−0.01; 0.15) |
| Model 1c | Emotional IPV | −0.02 | (−0.08; 0.03) | −0.02 | (−0.07; 0.03) |
| Model 1d | Any IPV | 0.04 | (−0.01; 0.09) | 0.04 | (−0.01; 0.09) |

Notes: Analyses were conducted separately for each component of IPV. The 95% confidence intervals (CIs) of the unadjusted coefficients and adjusted coefficients (aCoef.) are in parentheses. Bolding indicates $p < 0.05$.

(a) Adjusted for the following covariates: age, employment status, couple's educational attainment, parity, wealth score tercile, religion, and region of residence.

*** $p < 0.01$; ** $p < 0.05$.

planning [aCoef = −0.02, 95% CI = −0.07–0.03]. Similar analyses were carried out using a global composite variable that integrated all three dimensions of IPV; however, no relationship was identified between IPV as a collective measure and unmet need for family planning across any of the models employed (Table 3).

The four models outlined in Table 4 include interaction terms that explore the connection between IPV and reproductive coercion. Notably, the only significant interaction identified pertains to the relationship between the experiences of sexual IPV and reproductive coercion, with aCoef = 0.24 [95% CI = 0.03–0.44]. The predicted probabilities of unmet need for family planning as influenced by the variables of sexual IPV and reproductive coercion are shown in Fig 2. Among women in relationships who have experienced sexual IPV from their partners, the probability of facing an unmet need for family planning is considerably greater when reproductive coercion is present than when it is absent. Specifically, the likelihood of having an unmet need increases to approximately 50% in instances where sexual IPV occurs alongside reproductive coercion compared to approximately 20% when reproductive coercion is not involved.

The results of the different multivariate models assessing the association between IPV and unmet need for family planning are presented in Fig 3.

## Discussion

In this study, we examined how unmet needs for birth spacing and limitations among women of childbearing age in unions are associated with IPV in the provinces of Kinshasa and Kongo Central in the DRC. The present study contributes to the paucity of literature about IPV and its correlation with the unmet need for family planning in the SSA, and it is a pioneering study for the DRC, as previous studies focused on other SSA national contexts or used SSA pooled data. To the best of our knowledge, this is also the first study in the DRC as well as the rest of the SSA to explore the mechanisms of this association by considering reproductive control. Most existing studies rely on DHS data, which do not collect information on reproductive coercion, limiting their ability to explore its role in shaping the unmet need for family planning.

Our analysis introduces several other noteworthy aspects compared with previous studies on IPV and reproductive health in the DRC. First, we used a more comprehensive approach to IPV by considering three components (sexual, physical, and emotional), whereas most previous studies focused solely on sexual IPV or a combination of sexual and

**Table 4. Linear probability models of the association between IPV and unmet need for family planning (n = 1,382). With interaction term.**

| Model | Variable | aCoef. (a) | 95% CI |
|---|---|---|---|
| Model 2a | Sexual IPV | 0.04 | (−0.5; 0.13) |
| | Reproductive coercion | 0.01 | (−0.08; 0.11) |
| | Sexual IPV * Rep. Coercion | **0.24**** | **(0.03; 0.44)** |
| Model 2b | Physical IPV | 0.05 | (−0.03; 0.13) |
| | Reproductive coercion | 0.09 | (−0.02; 0.20) |
| | Physical IPV* Rep. Coercion | 0.02 | (−0.18; 0.22) |
| Model 2c | Emotional IPV | −0.03 | (−0.10; 0.02) |
| | Reproductive coercion | 0.09 | (−0.02; 0.21) |
| | Emotional IPV* Rep. Coercion | 0.04 | (−0.14; 0.22) |
| Model 2d | Any IPV | 0.01 | (−0.05; 0.06) |
| | Reproductive coercion | 0.02 | (−0.11; 0.15) |
| | Any IPV* Rep. Coercion | 0.14 | (−0.04; 0.32) |

**Notes**: Analyses were conducted separately for each component of IPV. The 95% confidence intervals (CIs) of the adjusted coefficients (aCoef.) are in parentheses. Ref.: reference. Bolding indicates p < 0.05.

(a) Adjusted for the following covariates: age, employment status, couple's educational attainment, parity, wealth score tertile, religion, and region of residence.

*** p < 0.01; ** p < 0.05.

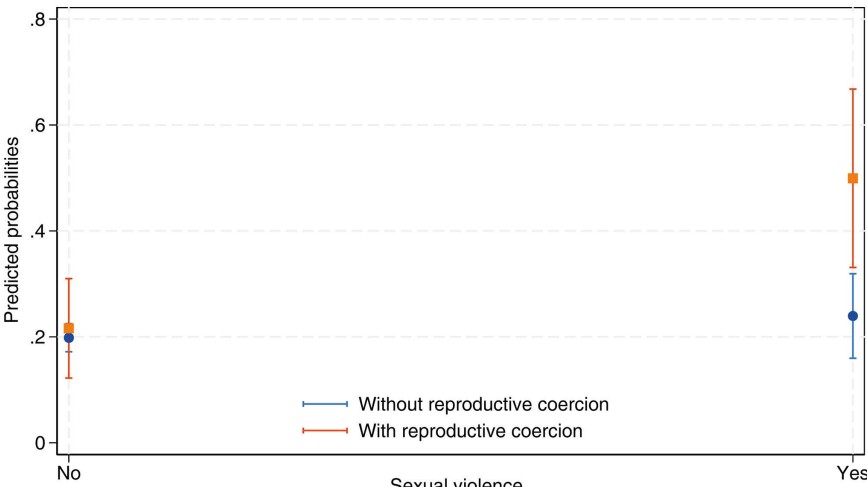

**Fig 2. Predicted probabilities of unmet need.** Interaction between sexual IPV and reproductive coercion.

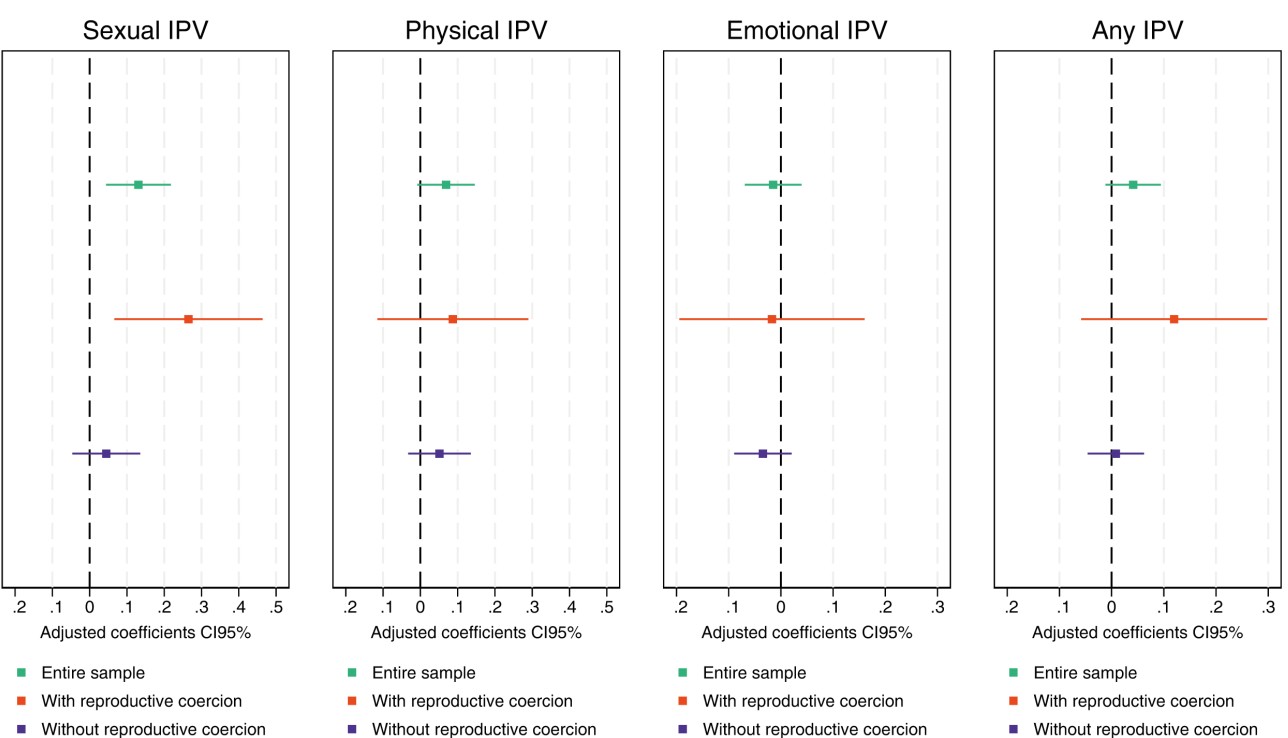

**Fig 3. Linear probability models of the association between IPV and unmet need by form of IPV and stratified by exposure to reproductive coercion.**

physical IPV [56]. Second, our study leverages the new PMA dataset [51], which includes information on IPV, albeit limited to two provinces. The last major study incorporating an IPV component in the case of the DRC was the 2013–2014 DHS and, given that IPV prevalence and its impact can evolve over time, updating this evidence is crucial. Third, unlike most

prior research that examines current contraceptive use, our study focuses on unmet need for family planning as our outcome variable. This variable encompasses current contraceptive use, unintended pregnancy, and the planning status of the last pregnancy or birth.

Our analyses reveal that IPV is still a widespread problem in the DRC, with emotional IPV being the most frequently reported form compared with physical and sexual IPV. Our results on the prevalence of IPV were similar to those of the recent 2023–2024 DRC DHS and showed a downward trend over the past decade [49], which could be attributed to changes in gender relations. In contrast, we reported a markedly lower prevalence of unmet need for family planning than for the 2023–2024 DHS, particularly in Kinshasa [50].

Several empirical studies have shown that women's lack of autonomy in abusive relationships restricts their access to contraception and family planning [30,31]. In response to our main research question, the results of our analyses broadly confirm the existence of a relationship between experience of IPV and unmet need for family planning in the DRC [18,57,58]. These results only partially support our first hypothesis (H1), as a significant association between unmet need for family planning and IPV was found only in the case of exposure to sexual IPV, whereas experience of physical and emotional IPV was not associated with unmet need. These results contrast with research conducted elsewhere, which has shown a higher likelihood of contraceptive use among victims of physical IPV [41,59] and emotional IPV [19,22,35]. However, these findings are consistent with previous literature from Côte d'Ivoire and Burkina Faso [60], the Maldives, Nepal, Pakistan, and India [20], and the Philippines [41], which reported no effect of physical and/or emotional IPV. The diversity of sampling approaches and contextual factors may help explain these divergent findings. For instance, in the DRC, more than half of women consider wife beating acceptable under certain circumstances [50]. This attitude toward the acceptability of IPV is not homogeneous, and substantial contextual and cultural differences exist both within and across countries. Future research using larger samples and including other regions of the DRC could validate the generalizability of the findings.

With regard to the remaining explanatory variables, the results of the descriptive and multivariate analyses confirmed findings from earlier studies. Women from the lowest wealth tercile [61], those who have not worked for pay in the past 12 months [62], younger women [12], those with more children [12,63], those living in low educational homogamy [64], women who were more educated than their partners, and those from the Kongo Central region [44] have higher levels of unmet need for family planning. In addition, high levels of unmet need were observed among women of other religious denominations and among those with no religious affiliation compared with Christian women.

In our second hypothesis (H2), we posit that men might exert violent control over women's reproductive choices by sabotaging contraceptive use and forcing women to become pregnant. In accordance with a previous study conducted in developed countries, we demonstrated that IPV, particularly sexual IPV, is associated with instances of reproductive coercion [42]. Our estimates of the interaction terms indicate that the combined effect of the experience of sexual IPV and reproductive control increases the probability of an unmet need for family planning more than the experience of sexual IPV alone without reproductive control does [42]. This interaction was confirmed only for sexual IPV and not for physical and emotional IPV; thus, our second hypothesis is partially confirmed.

## Strengths and limitations

Facilitated by recent PMA data, to the best of our knowledge, this is the first study of the association between IPV and unmet need for family planning, specifically for the DRC, and the first study in SSA to examine the underlying mechanisms of the association by considering reproductive coercion. Compared with previous studies on the Global North [42], which relied on convenience samples drawn from a limited number of family planning clinics, the present study benefited from a random sampling methodology and a larger sample size. In addition, the study made a significant contribution to the literature by considering several forms of IPV (sexual, physical, and emotional), also enabled by recent PMA data.

While the results of this study are original and striking, this study has several limitations. First, the main limitation lies in the cross-sectional nature of the data, resulting in a lack of detail on the chronology of events. We can determine an association between the two phenomena but not the existence of a cause-and-effect relationship. Second, underreporting is a major concern in research on IPV, particularly because of the sensitivity of the topic and concerns about participants' privacy and safety [65]. To address these issues, the PMA protocol requires strict confidentiality during interviews, and data collection is conducted by female interviewers to increase participants' confidence. Third, it is important to note that parity, which is considered a confounding factor, can also be influenced by sexual IPV, becoming an intermediate factor in the IPV-unmet need link. Finally, the results of this analysis pertain only to the situation in two provinces of the DRC, which does not allow for national generalization. Sociocultural specificities vary greatly across provinces, and some are characterized by limited access to health services.

## Implications

IPV is not only highly prevalent in the DRC but also positively associated with unmet need for family planning in the context of reproductive coercion. This association may imply bidirectional causality, and the interpretation of our findings is therefore twofold. First, IPV perpetrated by men may aim to enforce a pregnancy, thereby preventing women from accessing family planning services. This hypothesis was considered in most of previous studies [15,39,42,44]. In this case, women experience fear and inability to negotiate with their partner to space or stop subsequent pregnancies [15,44] and probably even to formulate their reproductive desires. For health care professionals, this implies that women with higher IPV risk are less likely to access family planning services. Community-level collaborators could help address this barrier by identifying situations where hegemonic masculinity manifests as violence, undermining women's reproductive autonomy.

A second and alternative interpretation of our findings refers to reverse causality, where a woman's desire to limit or space births may provoke a violent reaction from her partner. Qualitative research conducted in Kenya [43] supports this alternative assumption, showing that partners use IPV and/or reproductive coercion in response to women's use of contraception to limit births and may accuse them of engaging in prostitution or having extramarital affairs as the reason for taking contraceptive pills. Health care professionals play a key role in identifying situations of violence, and during antenatal care visits or when family planning services are offered, simple IPV screening instruments should be employed [66]. Confidentiality and privacy during consultations are vital to avoid violent reactions from the partner.

Interestingly, previous studies have shown that African men experience IPV perpetrated against women as an unpleasant reality but as necessary to maintain the established status quo rather than as an attempt to restrict women's rights [67]. These findings suggest that the importance of norms (the status quo) and the need to involve men in any program aiming to increase women's autonomy, their capacity to make decisions, and their ability to engage in negotiations regarding reproduction. Family planning programs including men could raise their awareness of the health risks for women with repeated and close births [68]. Additionally, and this specifically relates to our findings, men should be educated on the mechanisms of IPV and on how their reproductive expectations and coercive behaviors may escalate into violence with severe consequences for women's well-being.

Our study has included methodological implications. The specific underlying mechanisms behind the associations among IPV, reproductive coercion, and unmet need have received little or no attention in quantitative studies. Quantitative longitudinal studies could help clarify the sequence of events, including more complex sequences, with women who initially desired to limit and space births but later conformed to their partner's demands due to violence and coercion.

## Conclusion

The results of this research revealed a connection between IPV and unmet need for family planning in the DRC, supporting findings from earlier studies conducted in various other nations. This study identified sexual IPV as the only type

of IPV significantly associated with unmet family planning requirements, which is consistent with the majority of previous research focused on SSA [18,33,57]. Another key contribution of our study is the identification of the mechanisms that link IPV to unmet need for family planning. When sexual IPV is combined with reproductive coercion, the relationship with unmet need for family planning becomes even more pronounced.

## Supporting information

**S1 Fig. Sampling flow diagram.**
(TIF)

**S2 Table. Results of the Breusch–Pagan/Cook–Weisberg test for heteroskedasticity.**
(DOCX)

**S3 Table. Full size table of linear probability models for the association between IPV and unmet need for family planning among married or cohabiting women aged 15–49 years in the DRC (n = 1,382).**
(DOCX)

## Acknowledgments

We would like to thank the PMA project for providing access to the data on which the results of this study are based.

## Author contributions

**Conceptualization:** Didier Mbombo Ndombe, Ester Lucia Rizzi.

**Data curation:** Didier Mbombo Ndombe.

**Formal analysis:** Didier Mbombo Ndombe.

**Funding acquisition:** Didier Mbombo Ndombe, Ester Lucia Rizzi.

**Methodology:** Didier Mbombo Ndombe, Ester Lucia Rizzi.

**Project administration:** Didier Mbombo Ndombe, Ester Lucia Rizzi.

**Resources:** Didier Mbombo Ndombe, Ester Lucia Rizzi.

**Software:** Didier Mbombo Ndombe.

**Supervision:** Didier Mbombo Ndombe, Ester Lucia Rizzi.

**Validation:** Didier Mbombo Ndombe.

**Visualization:** Didier Mbombo Ndombe.

**Writing – original draft:** Didier Mbombo Ndombe, Ester Lucia Rizzi.

**Writing – review & editing:** Didier Mbombo Ndombe, Ester Lucia Rizzi.

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
