## [Decision Letter · Decision Letter 0]

26 Sep 2025

PONE-D-25-18133Intimate Partner Violence, ‘Reproductive Sabotage’ and Unmet Need for Family Planning in the Democratic Republic of Congo: Evidence from Performance Monitoring for Action DataPLOS ONE

Dear Dr. Ndombe,

Thank you for submitting your manuscript to PLOS ONE. After careful consideration, we feel that it has merit but does not fully meet PLOS ONE’s publication criteria as it currently stands. Therefore, we invite you to submit a revised version of the manuscript that addresses the points raised during the review process.

We look forward to receiving your revised manuscript.

Kind regards,

Vandana Dabla

Academic Editor

PLOS ONE

Journal Requirements:

1. Please ensure that your manuscript meets PLOS ONE’s style requirements, including those for file naming. The PLOS ONE style templates can be found at

3. Thank you for stating the following in your financial information:

[Université Catholique de Louvain              http://dx.doi.org/10.13039/100007370 /             Didier Mbombo Ndombe].

Reviewers' comments:

Reviewer’s Responses to Questions

**Comments to the Author**

1. Is the manuscript technically sound, and do the data support the conclusions?

Reviewer #1: Yes

Reviewer #2: Yes

Reviewer #3: No

2. Has the statistical analysis been performed appropriately and rigorously? 

Reviewer #1: Yes

Reviewer #2: Yes

Reviewer #3: No

3. Have the authors made all data underlying the findings in their manuscript fully available?

Reviewer #1: Yes

Reviewer #2: Yes

Reviewer #3: Yes

4. Is the manuscript presented in an intelligible fashion and written in standard English?

Reviewer #1: No

Reviewer #2: Yes

Reviewer #3: No

5. Review Comments to the Author

Reviewer #1: i personally acknowledge your efforts .i have attached the comprehensive report plz take your time and revise few things for the next .plz correct the language and the spellings according to standard of PLOS one.

i hope you may understand the query and will submit again well in time

regards

Reviewer #2: This is a well-structured study that examines the link between intimate partner violence and unmet need for family planning in the DRC, uniquely highlighting reproductive coercion as a key modifier, using PMA data, sound methodology and offers critical public health insights for integrated IPV and reproductive health interventions. Below are a few recommendations for improvement:

1. Statistical Transparency: While the use of linear probability models (LPM) is appropriately, the manuscript would benefit from explaining why interaction effects were only assessed for reproductive coercion and not for other potential effect modifiers such as education or wealth. This addition could help rule out alternative moderating factors. Furthermore, while the results of the Breusch-Pagan/Cook-Weisberg test for heteroskedasticity not presented. Reporting these findings either within the main text or as supplementary material would enhance methodological transparency.

2. Causal Interpretation: The authors appropriately note the limitation of the study’s cross-sectional design in precluding causal inference. While this is a typical limitation, the discussion would be enhanced by suggesting concrete approaches for future research such as longitudinal study designs or leveraging follow-up waves of PMA data to better examine causal relationships over time.

3. The manuscript briefly acknowledges the possibility that a woman’s intention to limit fertility may itself provoke IPV; however, this alternative hypothesis or causal pathway is insufficiently examined. Expanding the discussion to more fully consider the potential bidirectional nature of the relationship between IPV and fertility preferences would add valuable nuance and depth to the analysis.

4. Generalizability: While the study clearly notes its focus on Kinshasa and Kongo Central, the discussion would benefit from elaborating on how regional socio-cultural, economic, or health system differences might limit broader applicability across the DRC or SSA.

5. Under-reporting of IPV: Strengthen the discussion by outlining PMA’s measures to reduce under-reporting, such as interviewer training, use of private settings, and assurance of confidentiality, to enhance confidence in the reliability of self-reported IPV data.

6. Editorial Corrections: Ensure consistent use of terminology (e.g., “DRC” instead of “RDC”), correct minor typographical errors, and verify that all citations accurately reflect the studies they reference to maintain clarity and scholarly accuracy.

7. Contextualization of Findings: The authors should further explore why physical and emotional IPV were not associated with unmet need for family planning, in contrast to findings from other regions. Discussing possible cultural norms, reporting biases, differences in measurement, or the unique social dynamics of the DRC could offer important context and help explain these discrepancies for readers.

Reviewer #3: The central focus of this manuscript is to examine the association between domestic violence (binary, or multi-category variable) and unmet needs for family planning ( a well-defined binary variable/response), in a sample of study subjects collected within the Democratic Republic of Congo. The study objectives are on target. My comments, primarily statistical, are as follows:

(a) It’s not cloear to me why the authors chose to consider linear probability models (LPM) for their analysis, as alternatives to the well-explored logistic regression approach, now adjusted for survey weighting. The LPMs, in my opinion, are never appealing for modeling discrete (here, binary) response, given that they may encounter range issues, i.e., the predicted probability can be > 1, or < 0. A lot of software (such as, the R survey package) can easily conduct logistic regression, with survey weights, and the basis of going the LPM route is not clear.

(b) While constructing the primary independent variable ("domestic violence"), the three main variables (sexual, physical and emotional) were considered as dummy (1/0), and thereafter, a global variable was constructed after summing up the 3 binary components. It’s not clear if this "global variable" was used alongside the 3 dummy constructs in the same regression model. Such a route can induce inherent confounding.

(c) The discussion section should mention that the current findings are only based on the sample derived from the DRC region, and should allude to future studies with larger sample sizes across various regions to study the generalizability of the association measure.

6. PLOS authors have the option to publish the peer review history of their article (what does this mean? ). If published, this will include your full peer review and any attached files.

**Do you want your identity to be public for this peer review?** For information about this choice, including consent withdrawal, please see our Privacy Policy .

Reviewer #1: **Yes:** Dr ASAD ABRAR

Reviewer #2: **Yes:** Seth Kofi Abrokwa

Reviewer #3: No

---

## [Author Response · Author response to Decision Letter 1]

2 Dec 2025

Peer review report - Repartition of articles tasks for revision

Journal Requirements:

1. Please ensure that your manuscript meets PLOS ONE’s style requirements, including those for file naming. The PLOS ONE style templates can be found at

3. Thank you for stating the following in your financial information:

[Université Catholique de Louvain http://dx.doi.org/10.13039/100007370 /Didier Mbombo Ndombe].

Response: The funders had no role in study design, data collection and analysis, decision to publish, or preparation of the manuscript.

5. Review Comments to the Author

Reviewer 1**************************

Reviewer #1: I personally acknowledge your efforts. I have attached the comprehensive report plz take your time and revise few things for the next, plz correct the language and the spellings according to standard of PLOS one. I hope you may understand the query and will submit again well in time

regards

The study addresses a highly important topic, and overall the manuscript is well written. However, some sections require minor revisions to enhance clarity, coherence, and overall presentation, making the article more compelling and polished.

Abstract

1. Short the abstract as per the journal’s guidelines (maximum limit of 300 words).

Response: Thank you, we have shortened the abstract to 300 words.

2. In the abstract, the abbreviation ‘DRC’ is used for the Democratic Republic of Congo, while in the conclusion, ‘RDC’ appears to refer to the same country. Please ensure consistency in abbreviation throughout the manuscript to avoid confusion.

Response: Thank you for this remark. The correct abbreviation is DRC and is now consistently used.

Introduction

1. The title indicates that the study addresses both ‘reproductive sabotage’ and unmet need for family planning. However, the concept of ‘reproductive sabotage’ is not defined or discussed in the introduction or elsewhere in the manuscript. Please provide a clear explanation and contextual background of this term to align with the study’s focus.

Response: Thank you for this remark. The expression ‘reproductive sabotage’ has been omitted from the title. The present title is: “Intimate Partner Violence and Unmet Need for Family Planning in the Democratic Republic of Congo: The Moderating Role of Reproductive Coercion”.

2. In lines 105–106, a positive association between domestic violence and unmet need for family planning in sub-Saharan Africa (SSA) is reported without noting any exceptions. However, in lines 107–108, a negative association is mentioned. This contradiction should be clarified—please explain the context or differences in study findings to avoid confusion.

Response: We have added clarifications to the text. The first study (lines 105–106) used pooled data from 26 SSA countries, whereas the second study relied on pooled data from only six SSA countries (Cameroon, Kenya, Malawi, Rwanda, Uganda, and Zimbabwe).

Materials and Methods

1. What is PMA in line 149? Please avoid using abbreviations without first defining them. Write out the full term followed by the abbreviation in parentheses upon first use, and then use the abbreviation consistently in subsequent sections.

Response: The PMA mentioned in line 149 refers to Performance Monitoring for Action. It is already defined in the abstract, on line 25. We have redefined it in the data source section.

2. Clearly explain how sample size was calculated.

Response: We have added a sampling flow diagram as supplementary material and included the following text to explain how the sample size was calculated: “The sampling flow diagram is provided in S1 Fig.” We have also added further details regarding the sample design used in the PMA project.

We have added the following text: “The sample was designed to provide province level estimates of modern contraceptive prevalence rate among all women, with a margin-of-error of 5% at the province level. In each cluster, 35 households were randomly selected for interview, and all women aged 15–49 years in the selected households were eligible for the survey”.

3. Provide reference number of the ethical approval issued by the institute.

Response: The PMA project received ethical approval from the Institutional Review Boards (IRBs) at the Johns Hopkins Bloomberg School of Public Health (#14590) in November 2021 and the Kinshasa School of Public Health (KSPH) (#ESP/CE/159B/2021) in October 2021.

We have provided this information in the ethical approval section.

Results. Lines 302–303 include interpretation of the results, which should be reserved for the discussion section. Kindly relocate this interpretation to the discussion to maintain structural clarity.

Response: We have relocated this text to the discussion.

Discussion. Interpret the results in the discussion section and elaborate further by integrating recent findings from other lower-middle-income countries to provide broader context and support for the study’s focus.

Response: Thank you. We have relocated to the Discussion section the portions of the Results that were previously discussed at the end of the Results section. We have also added recent findings from lower-middle-income countries and/or sub-Saharan Africa that relate to our study, in order to provide broader context.

Conclusion. The conclusion should be concise, summarizing the key findings clearly and providing actionable recommendations based on the study results. Please consider shortening it to enhance focus and clarity. Several grammatical and spelling errors were noted throughout the manuscript. These should be carefully reviewed and corrected to improve the overall readability and professionalism of the article.

Response: Thank you for this remark. We have shortened the conclusion and corrected the grammatical and spelling errors noted throughout the manuscript.

Reviewer 2**************************

Reviewer #2: This is a well-structured study that examines the link between intimate partner violence and unmet need for family planning in the DRC, uniquely highlighting reproductive coercion as a key modifier, using PMA data, sound methodology and offers critical public health insights for integrated IPV and reproductive health interventions. Below are a few recommendations for improvement:

1. Statistical Transparency: While the use of linear probability models (LPM) is appropriately, the manuscript would benefit from explaining why interaction effects were only assessed for reproductive coercion and not for other potential effect modifiers such as education or wealth. This addition could help rule out alternative moderating factors. Furthermore, while the results of the Breusch-Pagan/Cook-Weisberg test for heteroskedasticity not presented. Reporting these findings either within the main text or as supplementary material would enhance methodological transparency.

Response: Thank you for the comment. We did not assess interactions for other potential effect modifiers because this was not the objective of the analysis. The analysis was designed to test the interaction only for reproductive coercion. Moreover, we tested the interactions with these two variables out of curiosity; however, the results were not statistically significant.

We have added the results of the Breusch-Pagan/Cook-Weisberg test for heteroskedasticity as supplementary material.

2. Causal Interpretation: The authors appropriately note the limitation of the study’s cross-sectional design in precluding causal inference. While this is a typical limitation, the discussion would be enhanced by suggesting concrete approaches for future research such as longitudinal study designs or leveraging follow-up waves of PMA data to better examine causal relationships over time.

Response: We believe that we have already addressed this aspect in lines 476–478, where we suggest that future research should use a longitudinal study design to more appropriately address causal inference.

3. The manuscript briefly acknowledges the possibility that a woman’s intention to limit fertility may itself provoke IPV; however, this alternative hypothesis or causal pathway is insufficiently examined. Expanding the discussion to more fully consider the potential bidirectional nature of the relationship between IPV and fertility preferences would add valuable nuance and depth to the analysis.

Response: We have clarified and provided references regarding the potential bidirectional nature of the relationship between IPV and fertility preferences.

Qualitative research conducted in Kenya [44] supports this alternative assumption. The results showed that partners used IPV and/or reproductive coercion in response to women’s use of contraception to limit births. Some partners accused women of engaging in prostitution or having extramarital affairs as the reason for taking contraceptive pills.

4. Generalizability: While the study clearly notes its focus on Kinshasa and Kongo Central, the discussion would benefit from elaborating on how regional socio-cultural, economic, or health system differences might limit broader applicability across the DRC or SSA.

Response: We have provided some explanations for why the results do not allow for national generalization. For example, sociocultural specificities vary greatly across provinces, and some are characterized by limited accessibility to health services.

5. Under-reporting of IPV: Strengthen the discussion by outlining PMA’s measures to reduce under-reporting, such as interviewer training, use of private settings, and assurance of confidentiality, to enhance confidence in the reliability of self-reported IPV data.

Response: We have provided the measures that PMA implemented to build trust with participants during interviews and data collection.

6. Editorial Corrections: Ensure consistent use of terminology (e.g., “DRC” instead of “RDC”), correct minor typographical errors, and verify that all citations accurately reflect the studies they reference to maintain clarity and scholarly accuracy.

Response: We have corrected the grammatical and spelling errors noted throughout the manuscript.

7. Contextualization of Findings: The authors should further explore why physical and emotional IPV were not associated with unmet need for family planning, in contrast to findings from other regions. Discussing possible cultural norms, reporting biases, differences in measurement, or the unique social dynamics of the DRC could offer important context and help explain these discrepancies for readers.

Response: We have added this text to clarify this contrast: “Varied sampling strategies and contextual factors may help explain these divergent findings. For instance, in the DRC, more than half of women consider wife-beating acceptable under certain circumstances. However, this attitude toward the acceptability of violence is not homogeneous, and substantial contextual and cultural differences exist both within and across countries”.

Reviewer 3**************************

Reviewer #3: The central focus of this manuscript is to examine the association between domestic violence (binary, or multi-category variable) and unmet needs for family planning (a well-defined binary variable/response), in a sample of study subjects collected within the Democratic Republic of Congo. The study objectives are on target. My comments, primarily statistical, are as follows:

(a) It’s not clear to me why the authors chose to consider linear probability models (LPM) for their analysis, as alternatives to the well-explored logistic regression approach, now adjusted for survey weighting. The LPMs, in my opinion, are never appealing for modeling discrete (here, binary) response, given that they may encounter range issues, i.e., the predicted probability can be > 1, or < 0. A lot of software (such as, the R survey package) can easily conduct logistic regression, with survey weights, and the basis of going the LPM route is not clear.

Response: We justified the choice of this model by adding the reference to Mood (2010), which compares estimates from several models, including logistic models and linear probability models (LPMs). This article highlights that logistic regression estimates do not behave in the same way as linear regression estimates, as they are affected by omitted variables—even when those variables are not related to the independent variables included in the model. First, it has been demonstrated that the interpretation of odds ratios can be biased due to unobserved heterogeneity, which includes control variables that are unmeasured, measurable but not included, or simply omitted from the model. Second, comparing odds ratios across models that contain different independent variables is problematic, because unobserved heterogeneity may vary from one model to another. Third, comparing odds ratios across samples, groups within samples, or over time is also problematic, even when the models use the same independent variables, since unobserved heterogeneity may differ across samples, groups, or time periods.

We acknowledge that one of the limitations of the LPM is that predicted probabilities may exceed 1 or fall below 0, which is theoretically impossible. However, Mood (2010) emphasizes that this is not usually a major concern, especially if compared to disadvantages mentioned above.

The LPM was therefore preferred because of its ability to address issues encountered with the logistic model, particularly the problem of unobserved heterogeneity when comparing several models [55].

[55] Mood C. Logistic Regression: Why We Cannot Do What We Think We Can Do, and What We Can Do About It. European Sociological Review. 2010;26: 67–82. doi:10.1093/esr/jcp006.

(b) While constructing the primary independent variable ("domestic violence"), the three main variables (sexual, physical and emotional) were considered as dummy (1/0), and thereafter, a global variable was constructed after summing up the 3 binary components. It’s not clear if this "global variable" was used alongside the 3 dummy constructs in the same regression model. Such a route can induce inherent confounding.

Response: We confirm that we created a global variable (“any IPV”) by summing the three IPV components. However, this variable was not included in any model together with one of its individual components, since that component is already part of the global variable. In lines 344–347, we specified that Table 3 presents the results for each of these components, with each row corresponding to an independent model. Moreover, none of these variables appears in the list of covariates at the bo

---

## [Decision Letter · Decision Letter 1]

2 Feb 2026

PONE-D-25-18133R1Intimate Partner Violence and Unmet Need for Family Planning in the Democratic Republic of Congo: The Moderating Role of Reproductive CoercionPLOS One

Dear Dr. Ndombe,

Thank you for submitting your manuscript to PLOS ONE. After careful consideration, we feel that it has merit but does not fully meet PLOS ONE’s publication criteria as it currently stands. Therefore, we invite you to submit a revised version of the manuscript that addresses the points raised during the review process.

**ACADEMIC EDITOR:**  The quality  of written english is of concern and the author should pay attention to this.This should be done after the manuscript is revised before submission.2. The title of the manuscript should be rewritten to include the study design.3. Under the abstract, the study design should be clear.4. Under the discussion section, the strenght of the study should come before the limitation while unanswered questions or future studies should be placed before conclusion not under limitation of study as it is. Also, there should be a section on implication of findings to healthcare professionals and policy makers. This is different from recommendation.5. The conclusion should be based on the findings from this research and the stated objectives. Avoid overgeneralization and make it concise.6. the references are not consistent, pay attention to this especially the use of uppercase letters.    ==============================

We look forward to receiving your revised manuscript.

Kind regards,

Adewale Olufemi Ashimi, MBBS, MPH, PhD, FWACS

Academic Editor

PLOS One

Journal Requirements:

Reviewers' comments:

Reviewer’s Responses to Questions

**Comments to the Author**

1. If the authors have adequately addressed your comments raised in a previous round of review and you feel that this manuscript is now acceptable for publication, you may indicate that here to bypass the “Comments to the Author” section, enter your conflict of interest statement in the “Confidential to Editor” section, and submit your "Accept" recommendation.

Reviewer #1: All comments have been addressed

Reviewer #2: All comments have been addressed

Reviewer #3: All comments have been addressed

2. Is the manuscript technically sound, and do the data support the conclusions?

Reviewer #1: Partly

Reviewer #2: Yes

Reviewer #3: (No Response)

3. Has the statistical analysis been performed appropriately and rigorously? 

Reviewer #1: Yes

Reviewer #2: Yes

Reviewer #3: (No Response)

4. Have the authors made all data underlying the findings in their manuscript fully available?

Reviewer #1: Yes

Reviewer #2: Yes

Reviewer #3: (No Response)

5. Is the manuscript presented in an intelligible fashion and written in standard English?

Reviewer #1: Yes

Reviewer #2: Yes

Reviewer #3: (No Response)

6. Review Comments to the Author

Reviewer #1: (No Response)

Reviewer #2: I have reviewed the revised manuscript and the authors’ responses to the reviewers’ comments and have noted that the authors have adequately addressed my concerns, and I have no further comments at this time.

Reviewer #3: (No Response)

7. PLOS authors have the option to publish the peer review history of their article (what does this mean? ). If published, this will include your full peer review and any attached files.

**Do you want your identity to be public for this peer review?** For information about this choice, including consent withdrawal, please see our Privacy Policy .

Reviewer #1: No

Reviewer #2: No

Reviewer #3: No

---

## [Author Response · Author response to Decision Letter 2]

31 Mar 2026

RESPONSE TO THE ACADEMIC EDITOR:

1. The quality of written english is of concern and the author should pay attention to this. This should be done after the manuscript is revised before submission.

Response: We ensured that the English was edited by a professional service.

2. The title of the manuscript should be rewritten to include the study design

Response: The title was revised to explicitly reflect the study design, although we prefer to avoid using the expression “secondary analysis” here to simplify the title.

3. Under the abstract, the study design should be clear.

Response: The abstract was revised to explicitly reflect the study design, i.e., a secondary analysis of PMA data.

4.A. Under the discussion section, the strenght of the study should come before the limitation

Response: Done.

4.B. While unanswered questions or future studies should be placed before conclusion not under limitation of study as it is.

Response: Done.

4. C Also, there should be a section on implication of findings to healthcare professionals and policy makers. This is different from recommendation.

Response: We reorganized and rewrote the final sections to respond to this comment.

5. The conclusion should be based on the findings from this research and the stated objectives. Avoid overgeneralization and make it concise.

Response: Done.

6. the references are not consistent, pay attention to this especially the use of uppercase letters. Response: Done. The references are managed automatically by Zotero, and the issue arises from the journal’s style specifications, which require the use of uppercase letters. We attempted to modify this formatting manually.

REMARKS

Despite the revisions made to address the reviewers’ remarks, several areas still require further improvement. The following specific suggestions are recommended to enhance the clarity, accuracy, and overall quality of the manuscript:

• Language and Grammar: This manuscript requires improvement in English language usage and grammatical accuracy through thorough proofreading.

Response: We ensured that the English was edited by a professional service.

• Abbreviation Usage: The manuscript should ensure that each abbreviation is defined at its first occurrence in the text, after which the abbreviation may be used consistently in subsequent sections. Line 104 &117 check for IPV.

Response:

The article has been submitted for English editing before resubmission. We have also systematically reviewed and rectified all issues related to the use of abbreviations. With respect to violence, we chose to use the term “intimate partner violence” exclusively instead of “domestic violence” and have applied this expression consistently throughout the document.

• Novelty Claim & Clarity of Contribution: The manuscript’s claim of being the first study on domestic violence and unmet need for family planning in the DRC is overstated. The authors cite prior research from Congo addressing similar issues, which undermines the assertion of novelty. The paper should clarify how it differs from existing studies and highlights its unique contribution without overstating originality.

Response:

Thank you for this remark. We confirm that our study is the first in the DRC. However, we have rephrased the sentence in the introduction as follows: “To the best of our knowledge, this study is the first to examine the relationship between IPV and unmet need for family planning specifically in the DRC, as previous studies on SSA considered pooled data [18] or countries other than the DRC [22].”

• Methodology Link: The provided link for full details of the methodology (https://fr.pmadata.org/data/survey-methodology) is not accessible and should be corrected or replaced with a functional source.

Response:

We have incorporated the appropriate bibliographic citations and removed the original hyperlink.

• Sample Size Consistency: Information about the sample size is inconsistent with the details presented in S1 Fig.

Response:

Thank you for this remark. The previous figure contained an error: the correct number is 1,387. We have revised Fig. S1 to include an additional stage that clarifies how 1,382 observations were retained for the final analysis. In this step, we excluded 5 observations that lacked complete information on the variables included in our models.

• Study Variables: How authors differentiated between sexual violence and reproductive coercion? How reproductive coercion was assessed.

Response:

Both reproductive coercion and sexual violence are defined in the methodology section. Reproductive coercion refers to the exertion of control over women’s reproductive autonomy through coercive behaviors, including but not limited to sabotaging contraceptive use and pressuring women to become pregnant. Sexual violence is conceptualized as one dimension of intimate partner violence and encompasses both nonphysical and physical forms of coercion, including pressuring or insisting on sexual intercourse when it is not desired (without the use of physical force) and physically forcing an individual to engage in sexual intercourse.

• Study Setting and Population: More detailed information should be incorporated into the Methods section to provide better context and understanding.

Response:

We have included slightly detailed information in the Methods section. We modified the paragraph as follows: “For the purposes of this study, the analytical sample included women in unions residing in the DRC, specifically in the provinces of Kinshasa and Kongo Central, where the PMA project was being implemented. Kinshasa is a densely populated, predominantly urban capital with relatively better access to health services and information, while Kongo Central is distinguished by a substantial predominance of rural areas and faces significant inequalities in access to essential resources and services. With the objective of examining the association between intimate partner violence (IPV) and unmet need for family planning, the study population was restricted to women aged 15–49 years living in a union, that is, women who are at risk of pregnancy or presenting reproductive potential. In addition, the analyses were limited to women in unions who responded to the module on violence against women and girls to capture information on IPV. All other individuals were excluded from the analytical sample using the “marital status” variable.”

• Data Collection Process: The data collection process remains unclear. While the ethical approval section suggests primary data collection with training on confidentiality and informed consent, the Materials and Methods section describe the use of secondary data from the PMA database, creating inconsistency that should be clarified.

Response:

Thank you for this remark. In the abstract and “data source” section, we now stress that we are using secondary data from the PMA. We also removed the description of the ethical approval process. This section was requested by one of the reviewers, but it can create inconsistency. In the “data source” section, we now more synthetically state that the PMA research received ethical approval, and we added a reference for it.

• Statistical Analyses: Line 273 and 260 inconsistencies between thresholds of p value.

Response:

The level of statistical significance was set at p < 0.05. We have made the necessary corrections. Thank you.

• Results:

1. Line 304: The reported percentage of domestic violence is 32.5%. Please clarify how this figure was calculated and whether it is based on weighted or raw data.

Response:

We have clarified how the calculation was performed and have refined the column descriptions by explicitly specifying the denominators.

2. Lines 306–312: It is unclear whether these belong to the Results section or the Discussion. Kindly specify and ensure consistent section placement.

Response:

Thank you for noticing this redundancy. We maintained these results only in the Results section.

3. Table 1 (Line 313): The percentages for experienced reproductive coercion are reported as 11.9% and 11.4%, which appear inconsistent. Please verify and correct.

Response:

Thank you. We have systematically reviewed this section and corrected all the identified issues. The correct value is 11.4%.

4. Lines 332–333: Table 1 does not contain information on the prevalence of unmet need for family planning, yet these lines reference such data. Please reconcile this discrepancy.

Response:

Table 1 presents information on the prevalence of unmet need for family planning according to each category of the dependent variables. We added a note in Table 1 to clarify this point. The prevalence of unmet need is presented in Table 2. The tables are correctly cited in the text.

5. Lines 332–351: A table summarizing these results is missing. Consider adding the appropriate table for clarity and completeness.

Response:

Table 1 provides a comprehensive summary of these results. Columns N and % indicate the distribution of each covariate. The last column of Table 1 indicates the prevalence of unmet need for family planning according to each category of the dependent variable.

6. Table 2: The total N=294, but the sum of categories equals 298. This suggests a possible typographical or data-entry error. Please review and correct.

Response:

Thank you for noticing this error. We have systematically reviewed this section and corrected all the identified issues. The correct value is 298. We corrected Table 2 accordingly.

7. Line 356: Reference is made to “S3 Table,” but this table is not provided. Kindly include or clarify.

Response:

Table S3 is now provided as supplementary material; it contains comprehensive information pertaining to Models 1a, 1b, and 1c.

• Abstract: Line 30 define past years. How authors concluded this because last 12 months was used in the data analyses. Conclusion needs improvements.

Response:

Thank you for noticing this inconsistency. We used “past 12 months” throughout the document.

• General: The manuscript uses the terms domestic violence and sexual violence interchangeably, which creates conceptual confusion. Since the findings specifically pertain to sexual violence, the terminology should be standardized throughout the text to avoid misinterpretation. Furthermore, the title should be revised to explicitly reflect that the study focuses on sexual violence, ensuring consistency between the reported results and the manuscript’s framing.

Response:

Thank you for this comment. We elected to use the term intimate partner violence (IPV) exclusively rather than domestic violence. IPV encompasses three forms of violence: sexual IPV, physical IPV, and emotional IPV. We applied this terminology consistently throughout the document, including the title.

Implementing these improvements will help ensure the manuscript is more polished, precise, and aligned with scholarly standards.

women’s polygyny status

---

## [Decision Letter · Decision Letter 2]

13 Apr 2026

PONE-D-25-18133R2The Association between Intimate Partner Violence and Unmet Need for Family Planning in the Democratic Republic of the Congo: An Analysis of the Moderating Role of Reproductive Coercion based on PMA SurveyPLOS One

Dear Dr. Ndombe,

Thank you for submitting your manuscript to PLOS ONE. After careful consideration, we feel that it has merit but does not fully meet PLOS ONE’s publication criteria as it currently stands. Therefore, we invite you to submit a revised version of the manuscript that addresses the points raised during the review process.

The comment regarding the title of the manuscript is yet to be addressed.

We look forward to receiving your revised manuscript.

Kind regards,

Adewale Olufemi Ashimi, MBBS, MPH, PhD, FWACS

Academic Editor

PLOS One

Journal Requirements:

Reviewers' comments:

Reviewer’s Responses to Questions

**Comments to the Author**

1. If the authors have adequately addressed your comments raised in a previous round of review and you feel that this manuscript is now acceptable for publication, you may indicate that here to bypass the “Comments to the Author” section, enter your conflict of interest statement in the “Confidential to Editor” section, and submit your "Accept" recommendation.

Reviewer #1: All comments have been addressed

2. Is the manuscript technically sound, and do the data support the conclusions?

Reviewer #1: Yes

3. Has the statistical analysis been performed appropriately and rigorously? 

Reviewer #1: Yes

4. Have the authors made all data underlying the findings in their manuscript fully available?

Reviewer #1: Yes

5. Is the manuscript presented in an intelligible fashion and written in standard English?

Reviewer #1: Yes

6. Review Comments to the Author

Reviewer #1: All the comments have been addressed. Recommended as accepted. Authors improved the manuscript as per requirements.

7. PLOS authors have the option to publish the peer review history of their article (what does this mean? ). If published, this will include your full peer review and any attached files.

**Do you want your identity to be public for this peer review?** For information about this choice, including consent withdrawal, please see our Privacy Policy .

Reviewer #1: **Yes:** Dr. Asad Abrar

---

## [Author Response · Author response to Decision Letter 3]

15 Apr 2026

1. The title of the manuscript should be rewritten to include the study design

Response: The new proposed title is:

"Intimate Partner Violence and Unmet Need for Family Planning in the DRC: A Secondary Analysis of the Moderating Role of Reproductive Coercion using PMA Data"

We believe this version more accurately describes the methodology while maintaining the focus on our key findings.

We hope this revised title meets your expectations and remain at your disposal for any further information and suggestion.

---

## [Editor Report · Decision Letter 3]

16 Apr 2026

PONE-D-25-18133R3Intimate Partner Violence and Unmet Need for Family Planning in the Democratic Republic of the Congo: A Secondary Analysis of the Moderating Role of Reproductive Coercion using PMA DataPLOS One

Dear Dr. Ndombe,

Thank you for submitting your manuscript to PLOS ONE. After careful consideration, we feel that it has merit but does not fully meet PLOS ONE’s publication criteria as it currently stands. Therefore, we invite you to submit a revised version of the manuscript that addresses the points raised during the review process.

The title should not contain any abbreviations.

We look forward to receiving your revised manuscript.

Kind regards,

Adewale Olufemi Ashimi, MBBS, MPH, PhD, FWACS

Academic Editor

PLOS One
---

## [Author Response · Author response to Decision Letter 4]

17 Apr 2026

Response: As requested, we have removed the abbreviation from the title. The revised title is:

"Intimate Partner Violence and Unmet Need for Family Planning in the Democratic Republic of the Congo: A Secondary Analysis of the Moderating Role of Reproductive Coercion Using Performance Monitoring for Action Data"

We hope this revised title meets your expectations and remain at your disposal for any further information and suggestion.

---

## [Editor Report · Decision Letter 4]

20 Apr 2026

Intimate Partner Violence and Unmet Need for Family Planning in the Democratic Republic of the Congo: A Secondary Analysis of the Moderating Role of Reproductive Coercion using Performance Monitoring for Action Data

PONE-D-25-18133R4

Dear Dr. Ndombe,

We’re pleased to inform you that your manuscript has been judged scientifically suitable for publication and will be formally accepted for publication once it meets all outstanding technical requirements.

Kind regards,

Adewale Olufemi Ashimi, MBBS, MPH, PhD, FWACS

Academic Editor

PLOS One
---

## [Editor Report · Acceptance letter]

PONE-D-25-18133R4

PLOS One

Dear Dr. Ndombe,

I'm pleased to inform you that your manuscript has been deemed suitable for publication in PLOS One. Congratulations! Your manuscript is now being handed over to our production team.

Kind regards,

on behalf of

Dr. Adewale Olufemi Ashimi

Academic Editor

PLOS One